# Synthesis of CaFe_2_O_4_-NGO Nanocomposite for Effective Removal of Heavy Metal Ion and Photocatalytic Degradation of Organic Pollutants

**DOI:** 10.3390/nano11061471

**Published:** 2021-06-01

**Authors:** Manmeet Kaur, Manpreet Kaur, Dhanwinder Singh, Aderbal C. Oliveira, Vijayendra Kumar Garg, Virender K. Sharma

**Affiliations:** 1Department of Chemistry, Punjab Agricultural University, Ludhiana 141001, Punjab, India; manmeetgill885@gmail.com; 2Department of Soil Science, Punjab Agricultural University, Ludhiana 141001, Punjab, India; dhanwinder@pau.edu; 3Institute of Physics, University of Brasilia, Brasilia 70000-000, Brazil; aderbal47@gmail.com (A.C.O.); vijgarg@gmail.com (V.K.G.); 4Program for Environment and Sustainability, Department of Environmental and Occupational Health, School of Public Health, Texas A&M University (TAMU), College Station, TX 77843-1266, USA

**Keywords:** CaFe_2_O_4_-NGO, Pb(II), congo-red (CR), *p*-nitrophenol (PNP), synergistic adsorption and photocatalysis

## Abstract

This paper reports the successful synthesis of magnetic nanocomposite of calcium ferrite with nitrogen doped graphene oxide (CaFe_2_O_4_-NGO) for the effective removal of Pb(II) ions and photocatalytic degradation of congo red and *p*-nitrophenol. X-ray diffraction (XRD), Fourier transform infrared (FT-IR), transmission electron microscopy (TEM), and scanning electron microscopy-energy dispersive X-ray (SEM-EDX) techniques confirmed the presence of NGO and CaFe_2_O_4_ in the nanocomposite. The Mössbauer studies depicted the presence of paramagnetic doublet and sextet due to presence of CaFe_2_O_4_ NPs in the nanocomposite. The higher BET surface area in case of CaFe_2_O_4_-NGO (52.86 m^2^/g) as compared to CaFe_2_O_4_ NPs (23.45 m^2^/g) was ascribed to the effective modulation of surface in the presence of NGO. Adsorption followed the Langmuir model with maximum adsorption capacity of 780.5 mg/g for Pb(II) ions. Photoluminescence spectrum of nanocomposite displayed four-fold decrease in the intensity, as compared to ferrite NPs, thus confirming its high light capturing potential and enhanced photocatalytic activity. The presence of NGO in nanocomposite offered an excellent visible light driven photocatalytic performance. The quenching experiments supported ^●^OH and O_2_^●−^ radicals as the main reactive species involved in carrying out the catalytic system. The presence of Pb(II) had synergistic effect on photocatalytic degradation of pollutants. This study highlights the synthesis of CaFe_2_O_4_-NGO nanocomposite as an efficient adsorbent and photocatalyst for remediating pollutants.

## 1. Introduction

Exponential increase in the use of heavy metals, nitroaromatics, and dyes in different industrial processes has raised the concern of water pollution [1]. Among toxic metal ions, Pb(II) ions are considered as the priority hazardous substance polluting water [2]. The maximum permissible limit for Pb(II) ions in drinking water is 0.05 mg/L, as recommended by the United States Environmental Protection Agency (USEPA). Contamination of Pb(II) ions mainly arises from the effluent wastes from industries such as mining, galvanization, smelting, and battery manufacturing. Pb(II) poisoning causes risk of high blood pressure, kidney damage, abdominal cramps, and headache [3]. In addition to inorganic pollutants, dyes and nitro aromatics have attracted great attention as organic pollutants contaminating the water. Nitroaromatics such as *p*-nitrophenol (PNP) is used mainly to manufacture drugs, fungicides, dyes, and to darken leather. Acute toxicity of *p*-nitrophenol causes irritation to the eyes, skin, and respiratory tract. When ingested, it causes abdominal pain and vomiting. Because of harmful nature of nitrophenols, they are listed as toxic pollutants by the U.S. Environmental Protection Agency [4]. Similarly, organic dyes and pigments are highly hazardous and toxic to a number of organisms [5,6,7,8]. Congo red (CR), an anionic dye, is found mainly in paper, printing, textile, and plastic industries. It is carcinogenic and causes irritation to skin, eyes, and affects alimentary tract, genital, and respiratory systems. In order to detoxify the environment, it is important to explore the facile and effective methods to remove these pollutants from water.

The different remediation techniques such as adsorption, ion exchange, precipitation, membrane filtration and photocatalysis can be used for water decontamination [9,10]. Among these, adsorption and photocatalysis are considered as the prominent methods due to their ease of operation [11,12]. In the recent years, graphene and its modified forms such as N-doped graphene, graphene oxide (GO) and N-doped graphene oxide (NGO) have been used as efficient adsorbents for removing various contaminants from wastewater. GO is considered as a good candidate due to the presence of hydroxyl, carboxyl, and epoxy functional groups on its surface [13,14,15,16]. Nitrogen doping on the surface of GO enhances its surface area, thus making it more efficient for adsorption [17]. However, the separation of NGO from the solution after adsorption or photocatalysis limits its use in the adsorption process. This problem can be overcome by fabricating the magnetic nanocomposites of NGO with ferrite nanoparticles (NPs). NGO acts as an excellent matrix material for dispersion of NPs.

Many studies have been conducted utilizing ferrite NPs and carbon-based materials as efficient adsorbents for the removal of heavy metal ions and dyes. Zhou et al. evaluated the carbon nanotubes/CoFe_2_O_4_ magnetic hybrid material for the removal of tetra bromo bisphenol A (TBBPA) and Pb(II). This hybrid showed the maximum adsorption capacities of 42.5 mg/g and 140.1 mg/g for TBBPA and Pb(II), respectively [18]. Pirouz et al. reported the adsorption of Pb(II) (107.5 mg/g) by using anhydride functionalized CaFe_2_O_4_ NPs [19]. Kaur et al. reported the adsorption of Ni(II) (100.0 mg/g) and Pb(II) (143.0 mg/g) ions by using MgFe_2_O_4_-GO magnetic nanocomposite [20]. Kaur and Kaur evaluated the MgFe_2_O_4_-GO nanocomposite for photocatalytic degradation of methylene blue dye under visible light irradiation in which MgFe_2_O_4_-GOnanocomposite with 1:0.5 *w*/*w* ratio showed complete degradation in 30min [21]. Deb et al. studied CaFe_2_O_4_ NPs for efficient adsorption of congo red with the maximum removal of 99.0% in 60 min [22]. Iqbal et al. reported La- and Mn-co-doped bismuth ferrite/Ti_3_C_2_MXene nanocomposite for the degradation of congo red with the complete degradation of dye in 30 min [23]. Rangumagar et al. studied the nitrogen doped graphitic carbon material (NGC) containing titanium oxide for the removal of nitrophenols from water [24]. The 80.0% adsorption of PNP was achieved using NGC–TiO_2_ compared to 74.0% with the use of NGC adsorbent.

Among various NPs, magnesium ferrite (MgFe_2_O_4_) and calcium ferrite (CaFe_2_O_4_) NPs are considered as suitable adsorbents as they have a large number of surface-active sites, which strengthen their use in fabricating the nanocomposites for improved adsorption [25,26]. The presence of magnesium and calcium as divalent ion is advantageous over ferrites having toxic metal ions such as nickel, zinc, cadmium ions. Literature survey shows that although CaFe_2_O_4_ is reported as adsorbent and photocatalyst but, no work has been reported on the synthesis and evaluation of CaFe_2_O_4_-NGO nanocomposite for the removal of heavy metals and photocatalytic degradation of organic pollutants. The N-doping improves the surface area of GO and combining CaFe_2_O_4_ NPs with NGO can further improve adsorptive and photocatalytic properties of nanocomposite. The presence of NGO along with CaFe_2_O_4_ can prevent the aggregation of NPs and can improve the photocatalytic properties of pristine CaFe_2_O_4_ NPs by retarding the recombination of e^−^-h^+^ pair. Thus, this work focuses on synthesis and exploring the both adsorptive and photocatalytic properties of CaFe_2_O_4_-NGO nanocomposite.

The facile ultra-sonication approach was used to fabricate CaFe_2_O_4_-NGO nanocomposite. The morphological details of the synthesized NPs and nanocomposite were explored using SEM-EDX and TEM analysis. The effect of structural variation on Mössbauer parameters and surface area was also studied. The adsorptive and photocatalytic properties of nanocomposite were examined by using Pb(II) ions, *p*-nitrophenol and congo red as the model pollutants. The results were compared with pristine CaFe_2_O_4_ NPs under different experimental conditions (pH, adsorbent dose, contact time, temperature, and solution concentration). Adsorption isotherms, kinetic modeling, and thermodynamic parameters were examined to predict the mechanism of adsorption and photocatalysis. Statistical analysis of data was also carried out to validate the results of isotherm and kinetic studies. The studies were further extended for synergistic adsorption of heavy metal ion and photocatalytic degradation of organic pollutants and reusability of the nanocomposite.

## 2. Materials and Methods

All used chemicals viz. graphite powder, ortho-phosphoric acid, sulfuric acid, potassium permanganate, hydrochloric acid, ammonium hydroxide (30%), ferric nitrate, calcium nitrate, citric acid, hydrogen peroxide, lead nitrate, congo red (CR) dye, and *p*-nitrophenol (PNP) were of analytical reagent grades. Distilled water was used to prepare solutions of different concentrations.

### 2.1. Synthesis of N-GO

NGO was synthesized using a facile ultrasonication method. Briefly, 0.5 g of GO was dissolved in 20.0 mL of 25% NH_4_OH solution at 25 °C and sonicated for 15, 30, and 60 min. The prepared solution was centrifuged at 5000 rpm, followed by repeated washings with water. The obtained blackprecipitates were dried in an oven at 60 °C overnight to obtain N-GO as the final product.

### 2.2. Synthesis of CaFe_2_O_4_

The 2.0 mol of Fe(NO_3_)_3_·9H_2_O and 1.0 mol of Ca(NO_3_)_2_·4H_2_O were dissolved in 20.0 mL de-ionized water. To this aqueous solution, 2.2 mol of citric acid was added. The above solution was magnetically stirred at 60 °C to which NH_4_OH solution was added to have the desired pH 7.0 and the solution transformed into sol. After continuous stirring for 8.0 h, the sol changed into gel, which was dried at 100 °C for 12 h and its volume increased by 5 times. Finally, the dried gel was grounded and calcined at 300 °C for 3 h to obtain CaFe_2_O_4_ as the final product.

### 2.3. Synthesis of Nanocomposite of N-GO and CaFe_2_O_4_ NPs

Nanocomposites of N-GO with CaFe_2_O_4_NPs i.e., CaFe_2_O_4_-NGO were prepared in three different *w*/*w* ratios of 1:1, 1:2, and 2:1 by the sonication method. Briefly, N-GO and CaFe_2_O_4_ NPs were dissolved separately in different *w*/*w* ratio in 15 mL ethanol and sonicated for 30 min. The solution of CaFe_2_O_4_ NPs was added to the solution of N-GO and the ultrasonication was repeated for 1 h. The resultant mixture was centrifuged and washed with water and subsequently dried in an oven at 60 °C overnight to prepare CaFe_2_O_4_-NGO as the final product.

### 2.4. Characterization

Different characterization techniques used to study the structure, particle morphology, surface area, and magnetic properties along with the instrumental details are given in Table 1. FT-IR spectra of the samples were taken by making their pellet with KBr. The frequency was measured as a wave number in the range 400–4000 cm^−1^. SEM micrographs of the samples were captured by scanning the surface with the focused beam of electrons. Samples were mounted rigidly on the specimen holder, using a conductive adhesive. For the TEM analysis, the samples were dispersed in distilled water. TEM micrographs of the samples were captured when the beam of electrons was transmitted through the sample to form the image. In XRD, copper was used as a target material for diffraction with Cu Kα radiation having λ = 1.5418 Å operated at 45 kV and 40 mA, in the 2θ range from 5 to 65°. The XPS spectra were obtained by irradiating the sample with beam of X-ray, while measuring the kinetic energy and number of electrons that escape from top 0 to 10 nm of the material being analyzed. In BET analysis, the amount of gas adsorbed was measured by a continuous flow procedure. The samples were degassed at 200 °C for 12 h before taking the measurements. For VSM analysis, the sample was placed in sensing coils. Vibrating sample component was made to undergo sinusoidal motion i.e., mechanically vibrated. A magnetic field appeared around the sample and once the vibration began, the magnetization of the sample was analyzed according to the changes occurring in relation to the timing of movement. The Mössbauer spectra of all the powdered samples were recorded with a constant acceleration transducer (Wissel) coupled to ^57^Co in Rh matrix source with an initial activity of around 5 mCi in the standard transmission geometry. For the detection of 14.4 keV gamma rays proportional counter was used. The velocity calibration per channel, and determination of the zero channel with reference to iron for isomer shift was carried with a 1.9 mg per ^57^Fe cm^2^ iron foil, using a 256 multichannel analyzer. All the isomer shifts are reported with reference to iron. Fifty mg of samples were uniformly distributed in a circular Teflon cell. The recorded spectra were least square fitted using commercially available program ‘MOSSWIN’.

### 2.5. Adsorption, Kinetics, Thermodynamics, and Desorption Studies

The CaFe_2_O_4_:NGO nanocomposites were comparatively analyzed for their adsorption potential for Pb(II) ions (Appendix A). Nanocomposite with 1:2 ratio was preliminarily screened as the best adsorbent among the synthesized nanocomposites and was used for carrying out detailed analysis. For the adsorption studies, the standard stock solution of Pb(II) of concentration 1000 mg/L was prepared by dissolving 1.598 g of Pb(NO_3_)·4H_2_O in 1000 mL of double distilled water. Working solutions were prepared by dilution of stock solution using double distilled water. Each experiment of the prepared materials was repeated thrice. The mean of the triplicates and the standard deviation were calculated, followed by the determination of standard error using the following equation:
$$\mathrm{Standard}\;\mathrm{error}\;(S.\;E.)=\frac{\mathrm{Standard}\;\mathrm{deviation}}{\surd n}$$

where n = 3 represents repeated runs. The error bars were then added into Microsoft excel using custom error bar option with the values obtained with the above equation. The effect of pH was studied by adjusting the solution pH with 0.1 M NaOH and 0.1 M HCl. The optimum pH for adsorption of Pb(II) by CaFe_2_O_4_ and CaFe_2_O_4_-NGO was determined, whereas the effect of the adsorbent dose (0.1–2.0 g/L) was studied under the optimum pH. In order to determine the effect of the contact time, the study was conducted in the time range of 0 to 180 min under the optimized conditions. The effect of temperature was studied by performing the experiments at five different temperature regimes ranging from 288 K to 328 K. The adsorption experiments were carried out at 25 ± 1 °C and the solutions were shaken in an orbital shaker operated at 130 rpm. The values of thermodynamic parameters were also calculated by the equations given in Appendix A. Adsorption data was fitted to the Freundlich, Langmuir, Temkin and Dubinin-Radushkevich (D-R) adsorption isotherms (Appendix A). The kinetic data were analyzed by testing pseudo-first-order and pseudo-second-order models (details in Appendix A). For validation of different kinetic and isotherm models, Sum of Squared Error (SSE), Chi Square Error (χ^2^), ‘F’-statistics and Log Likelihood Error (G^2^) were calculated from the equations in Appendix A. Desorption studies were conducted using 1.0 g/L of adsorbent in 100 mL solution containing 5.0 mg/L of Pb(II) solution where the desorbing medium was 0.1 M HCl. The influence of common co-existing cations in the Pb-Cd-Zn-Ni system and anions (Cl^−^, SO_4_^2−^ and NO_3_^−^) using different salts of concentration 5.0 mg/L on Pb(II) adsorption was also investigated using 1.0 g/L of CaFe_2_O_4_-NGO nanocomposite.

### 2.6. Optical Properties and Photocatalytic Activity Measurements

The important parameter of a semiconductor is its band gap (E_g_) i.e., the energy difference between the top of the valence band and the bottom of the conduction band. The band gap energy calculation involves the extrapolation of linear regime of the Tauc plot obtained by plotting E_g_ vs. (αhν)^2^ to perpendicularly cut the energy axis where ‘α’ is absorption coefficient and ‘E_g_’ of the synthesized photocatalysts was determined according to the following equation:
$$E_{g}=h\upsilon =\frac{\mathrm{hc}}{\lambda }=\frac{1240}{\lambda }.$$


The synthesized NPs and nanocomposite were also tested for fluorescence studies and the photo luminescence emission spectra were recorded in the wavelength range 400–800 nm.

The standard stock solutions of congo red and *p*-nitrophenol were prepared in 1000 mL distilled water. Working solutions were prepared by dilution of the stock solution with double distilled water. The effect of pH on the photodegradation of organic pollutants was examined, where the pH was adjusted between 1.0 and 9.0 with 0.1 M HCl and 0.1 M NaOH solution. In order to study the effect of dose on photodegradation, different doses of photocatalyst (0.01–2.0 g/L) were added into flasks each containing 100 mL solution of 2.0 mg/L of *p*-nitrophenol and congo red, separately under the optimized conditions of pH. The CaFe_2_O_4_-NGO nanocomposite was evaluated for the photocatalytic degradation of organic contaminants under visible light illumination by using light emitting diode of 60 W power. The solutions of different initial concentrations were prepared from the stock solutions and pH was adjusted to 3.0 for *p*-nitrophenol and congo red. For photodegradation of *p*-nitrophenol, 0.01 g of photocatalyst was added in 100 mL of 2.0 mg/L of solutions. Similarly, 0.1 g of nanocomposite was dispersed in 100 mL of 2.0 mg/L aqueous solution of congo red. The suspensions were kept in incubator shaker at 130 rpm for 30 min in order to maintain adsorption-desorption equilibrium. The solutions were subjected to the visible light irradiation after the addition of 0.2 mL of H_2_O_2_. After regular intervals of time, the appropriate aliquots of solutions were withdrawn, centrifuged, and the supernatant solutions were analyzed for change in the *p*-nitrophenol and congo red concentration spectrophotometrically at λ_max_ of 398 nm and 495 nm, respectively. Pseudo-first order and Pseudo-second order kinetic models were applied to elucidate the mechanism. For the validation of models, Sum of Squared Error (SSE), Chi Square Error (χ^2^), and ‘F’-statistic were calculated. The details of the kinetic models and statistical parameters are provided in Appendix A.

The effect of scavengers on photocatalytic degradation of congo red and *p*-nitrophenol was also studied. One mL aqueous solution of disodium ethylene diamine tetraacetate (10 mM) and methanol (holes scavenger), ascorbic acid (superoxide scavenger) and butanol (hydroxyl scavenger) were added separately to each flask containing congo red and *p*-nitrophenol. The solutions of congo red and *p*-nitrophenol without adding scavengers were used as controls. These solutions were subjected to visible light irradiation for 120 min for photocatalysis followed by analysis using spectrophotometer.

The recyclability of the CaFe_2_O_4_-NGO nanocomposite was evaluated by carrying out six successive cycles of photodegradation. In this experiment, 0.1 g and 0.01 g of nanocomposite was added in 100 mL solutions of congo red and *p*-nitrophenol, respectively. The solutions were shaken for 2 h at 130 rpm in order to attain adsorption-desorption equilibrium and were subjected to visible light irradiation. After 2 h, the samples were centrifuged and the centrifugates were analyzed for any change in concentration. After the completion of each cycle, the photocatalyst was washed with deionized water several times. The nanocomposite was collected by centrifugation and dried at 60 °C in an oven for next cycle of photocatalysis.

## 3. Results and Discussion

### 3.1. Structural Analysis

The phase purity and the crystal structure of synthesized NPs and nanocomposite were examined using X-ray diffraction (XRD) patterns as shown in Figure 1. The CaFe_2_O_4_ NPs showed the distinctive peaks at 2θ = 29.53°, 30.15°, 34.88°, 42.55° and 52.54° with hkl values of 220, 230, 320, 311, and 401, respectively suggesting the presence of crystallographic planes of CaFe_2_O_4_ NPs. All the peaks and Miller indices (hkl) of the face centered cubic (fcc) lattice observed in the patterns matched with JCPDS Data Card No. 50-1746 [27]. The XRD pattern of CaFe_2_O_4_-NGO nanocomposite showed the additional peak at 2θ = 11.55° (hkl = 001), which supports the presence of both N-GO and CaFe_2_O_4_ NPs in the nanocomposite.

The FT-IR studies were performed to know the presence of different functional groups on the surface of CaFe_2_O_4_ NPs and CaFe_2_O_4_-NGO nanocomposite. The bands for CaFe_2_O_4_-NGO indicated the presence of both components [28,29,30]. The details are given in Appendix A.

The SEM micrographs using Energy dispersive X-ray (EDX) were recorded in order to examine the surface topography of synthesized NPs and nanocomposite. The SEM image of CaFe_2_O_4_ NPs displayed the agglomerated particles due to the magnetic dipole interaction between ferrite NPs (Figure 2). Similar results were reported by Sulaiman et al. for superparamagnetic calcium ferrite NPs prepared using a sol-gel method [31]. The EDX mapping depicted the presence of Ca, Fe, and O contents (Figure 2 inset). The SEM image of CaFe_2_O_4_-NGO nanocomposite showed rough surface due to presence of CaFe_2_O_4_ NPs on the surface of N-GO and EDX mapping depicted the presence of N and C along with O, Fe, and Ca, thus authenticating the presence of N-GO in nanocomposite. The size distribution and the particle morphology of synthesized NPs and nanocomposite were determined by the TEM analysis. The TEM image of CaFe_2_O_4_ NPs displayed the agglomeration due to magnetic nature (Figure 3a) and cluster of particles with size ranging from 10 to 20 nm (Figure 3b). The TEM image of pristine GO showed layered structure whereas, N-GO had increased micro wrinkling due to nitrogen doping in GO (Figure 3a). The TEM image of CaFe_2_O_4_-NGO nanocomposite depicted the presence of wrinkled sheets of N-GO over which CaFe_2_O_4_ NPs were distributed (Figure 3c).

In order to study the surface properties of synthesized NPs and nanocomposite, the N_2_ adsorption-desorption isotherm measurements were conducted. Figure 4 displays the low-pressure nitrogen-gas-sorption isotherms along with pore size distribution (inset). The BET surface area and pore volume of the synthesized nanocatalysts are listed in Appendix A. The synthesized NPs and nanocomposite followed type IV isotherm of H3 type hysteresis loop, confirming their mesoporous nature. The BET surface area of CaFe_2_O_4_-NGO (52.86 m^2^/g) was higher than pristine CaFe_2_O_4_ NPs (23.45 m^2^/g). A similar trend was seen in measurements of pore volume. The higher surface area of CaFe_2_O_4_-NGO compared to CaFe_2_O_4_ NPs was attributed to the uniform distribution of NPs in nanocomposite, which prevented agglomeration of NPs. Mesoporosity of the nanocomposite increased as indicated by the pore volume, which resulted in higher BET surface area of nanocomposite.

The magnetic properties of the NPs and the nanocomposite were further characterized using VSM analysis with the applied field varying from −10 to +10 KO_e_. The NPs and nanocomposite displayed s-shaped narrow hysteresis loop signifying ferrimagnetic behavior (Figure 5a). The parameters of magnetic study for the samples are reported in Appendix A. The M_s_ value of CaFe_2_O_4_-NGO nanocomposite (2.38 emu/g) was lower than the pristine CaFe_2_O_4_ NPs (5.03 emu/g). This can be due to the addition of non-magnetic N-GO in the nanocomposite, which resulted in the decrease in magnetic permeability of the particles and quenching of magnetic moment. Thus, the magnetic nature of nanocomposite is advantageous for its magnetic separation after adsorption from the aqueous medium.

The Mössbauer spectra for CaFe_2_O_4_ NPs and CaFe_2_O_4_-NGO nanocomposite are shown in Figure 5b and the corresponding hyperfine parameters are given in Table 2. The Mössbauer spectrum of CaFe_2_O_4_ NPs displayed two sextets and a quadrupole doublet, corresponding to the different crystallographic sites occupied by iron with isomer shift (δ) and quadrupole splitting (Δ) values ranging from 0.258 to 0.340 mm/s and −0.047 to 0.691 mm/s, respectively. The estimated δ values are typical for octahedrally-coordinated Fe^3+^ in the high-spin state and Δ values were nearly identical to those reported by Tsipis et al. 2007 for CaFe_2_O_4_ NPs [32]. The range of quadrupole splitting in the present study reflect different values of the electric field gradients acting on Fe^3+^ nuclei in two non-equivalent octahedral positions of the pristine CaFe_2_O_4_ NPs. CaFe_2_O_4_-NGO NC exhibited sextet and a paramagnetic doublet with δ values ranging from 0.338 to 1.024mm/s and Δ values ranging from 0.047 to 0.675 mm/s, respectively. Here, the higher δ and Δ values for nanocomposite may result from the asymmetric charge distribution around Fe nuclei originating from the interaction with N-GO. Whereas, lesser ∆ values for pristine CaFe_2_O_4_ NPs reflected the presence of less distorted structure [33]. The doublet in the spectrum confirmed the presence of super paramagnetic fraction of NPs.

### 3.2. Optical Studies

The band gap energy of CaFe_2_O_4_NPs and CaFe_2_O_4_-NGO nanocomposite was calculated from UV-Visible Diffused Reflectance Spectra (DRS) and the results are shown in Figure 6a. Pristine CaFe_2_O_4_ NPs exhibited the optical band gap energy of 2.10 eV, which is in agreement with those reported in literature [34]. However, CaFe_2_O_4_-NGO nanocomposite had a narrow band gap energy of 1.92 eV. This decrease in band gap might be due to the appearance of new hybridized energy levels in the CaFe_2_O_4_-NGO nanocomposite. The interfacial contact between CaFe_2_O_4_ NPs and NGO resulted in the enhanced photocatalytic activity due to lengthening of the lifetime of photogenerated electron-hole pairs in the nanocatalyst. The additional localized states were developed in the CaFe_2_O_4_ NPs in the presence of N-GO that facilitated the separation of photo generated electrons and holes and efficiently suppressed the electron-hole pair combination. This phenomenon was not only related to the electronic transition between N-GO and CaFe_2_O_4_ NPs but also attributed to the inherent light absorption by N-GO. Thus, the formation of CaFe_2_O_4_-NGO nanocomposite resulted in excellent photocatalytic degradation of pollutants in the presence of visible light.

The photoluminescence emission spectra were also studied in the wavelength range of 400 to 800 nm for CaFe_2_O_4_NPs and CaFe_2_O_4_-NGO nanocomposite (Figure 6b). Emission peaks were observed at 400 and 700 nm due to the electronic transitions [35]. CaFe_2_O_4_-NGO nanocomposite displayed a four-fold decrease in luminous intensity due to the presence of NGO, thus exhibiting higher quenching efficiency over pristine CaFe_2_O_4_NPs. The intercalation of NGO sheets with CaFe_2_O_4_ NPs resulted in the decrease in peak intensity for CaFe_2_O_4_-NGO nanocomposite. This lowering in peak intensity could be correlated well with the band gap studies. The results further showed that the formation of CaFe_2_O_4_-NGO nanocomposite could slow down the photoinduced charge carrier recombination over CaFe_2_O_4_ NPs due to transfer of electrons from the conduction band of the photo-excited CaFe_2_O_4_ NPs to N-GO sheets and thus offer excellent potential for photocatalytic degradation of organic pollutants.

### 3.3. Adsorption Studies for Pb(II) Ions

#### 3.3.1. Removal Efficiency of CaFe_2_O_4_-NGO Nanocomposites

The removal (%) of Pb(II) ions at pH 6.0 using CaFe_2_O_4_-NGO nanocomposite was determined with varied ratios of CaFe_2_O_4_and N-GO i.e., 1:1, 1:2, and 2:1 (Appendix A). An amount of 0.01 g of CaFe_2_O_4_-NGO in a 1:2 ratio displayed higher percentage removal of 84.7 ± 2.2% for Pb(II) ions. A presence of greater amount of N-GO in nanocomposite with 1:2 ratio significantly lowered the agglomeration of the NPs and provided greater surface area for adsorption as compared to nanocomposites in the 1:1 and 2:1 ratio. Huong et al. reported a similar trend for functional manganese ferrite/graphene oxide nanocomposites for the adsorption of methylene blue dye and As(V) ions [36]. Thus, the results signify the role of NGO in increasing the adsorption potential of CaFe_2_O_4_-NGO nanocomposite in 1:2 ratio, which was used for further studies.

#### 3.3.2. Effect of Contact Time and Adsorption Kinetics

To estimate the adsorption rate for uptake of Pb(II) onto CaFe_2_O_4_ NPs and CaFe_2_O_4_-NGO nanocomposite, time dependent adsorption studies were conducted (Figure 7a). The adsorption rate was fast initially, and adsorption equilibrium was attained within 60 min for CaFe_2_O_4_ NPs and CaFe_2_O_4_-NGO, suggesting strong interactions between adsorbent and metal ions. CaFe_2_O_4_ NPs and CaFe_2_O_4_-NGO exhibited the percentage removal of 80.2 ± 0.7% and 88.6 ± 1.2% in 120 min, respectively.

In order to study the kinetics and the controlling mechanism involved in the adsorption process, pseudo-first order, and pseudo-second order kinetic models were used. Pseudo-first order equation assumed the interaction between one adsorbate molecule onto one active site on the surface of adsorbent while pseudo-second order model assumed interaction between one adsorbate molecule onto two active sites of the adsorbent [37]. For studying this model, the results were plotted between ln (q_e_-q_t_) and t (Figure 7b). The values for equilibrium adsorption capacities (q_e_) and rate constant (K_1_) calculated from the plots are reported in Table 3. There is a large difference between the experimental and calculated values of q_e_, indicating poor pseudo-first-order fit to experimental data (Table 3**)**. Pseudo-second order model was studied by plotting t/q_t_ vs. t (Figure 7c) and the values of rate constants determined from the plots are given in Table 3. The linearity of the plot and good agreement between the experimental and calculated q_e_ values, coefficient of determination (R^2^) approximately equal to 0.99 supported that the pseudo-second order model may explain the observed results. The fitting quality of the model to the experimental data was determined from the values of F- statistic, sum of squared error (SSE) and chi square (χ^2^) error. The values of these parameters are summarized in Table 3. Higher values of the F-statistics and the lesser values of SSE and χ^2^ error for pseudo-second order model as compared to pseudo-first order model supported better fitting of a former model.

#### 3.3.3. Effect of Adsorbent Dose and Temperature

The adsorbent dose studies displayed increase in removal efficiency with increase in adsorbent dose due to availability of more adsorption sites with increase in dosage (Appendix A). An optimum dose was determined to be 1.0 g/L. The details of adsorbent dose and thermodynamic studies are explained in Appendix A, respectively. The higher removal efficiency of nanocomposite over pristine NPs was correlated to the zeta potential studies (Appendix A), which depicted that more interactions occurred between negatively charged surface of nanocomposite and positively charged Pb(II) ions. The removal efficiency of Pb(II) ions increased with an increase in the temperature up to 328K, indicating endothermic nature of adsorption for mesoporous adsorbent. Moreover, the increased number of active adsorption sites, following the increase in temperature was observed due to the cleavage of some bonds near the active sites [38]. A similar trend was observed by Lingamdinne et al. for the adsorptive removal of Pb(II) and Cr(III) by using nickel ferrite-reduced graphene oxide nanocomposite [39]. A decrease at higher temperature was due to the desorption of adsorbed ions. The values of ∆H° and ∆S° were estimated using the slope and the intercept of the plot of lnK vs. 1/T (Appendix A) and the values of ∆G°, ∆H°, and ∆S° for CaFe_2_O_4_ NPs and CaFe_2_O_4_-NGO are given in Table 4.

The adsorption isotherm studies were performed to elucidate the underlying mechanism for the removal of Pb(II) ions by the synthesized adsorbents. Langmuir, Freundlich, Dubinin Radushkevitch, and Temkin adsorption isotherms were plotted for Pb(II) ion. The graph was plotted between 1/C_e_ vs. 1/q_e_ in order to study Langmuir adsorption isotherms and it yielded a straight line for the synthesized adsorbents, thus indicating that the Langmuir model was followed by CaFe_2_O_4_ NPs and CaFe_2_O_4_-NGO nanocomposite (Figure 8a). The adsorption parameters are presented in Table 5. The value of ‘q_m_’ i.e., maximum adsorbent uptake was found to be 520.0 mg/g and 780.5 mg/g for CaFe_2_O_4_ NPs and CaFe_2_O_4_-NGO, respectively. Whereas, the values of ‘b’ (Langmuir constant) were 0.014 L/mg, and 0.004 L/mg, respectively. The Freundlich plots (i.e., logq_e_ vs. logC_e_) also yielded linearity for both CaFe_2_O_4_ NPs and CaFe_2_O_4_-NGO nanocomposite (Figure 8b), and the estimated parameters are given in Table 5. The values of n and K_f_ were found to be 1.2 and 1.9, and 0.61 mg/L and 1.20 mg/L for CaFe_2_O_4_ NPs and CaFe_2_O_4_-NGO, respectively, which indicated the good adsorption capacity of the adsorbents towards Pb(II). The values of n>1 supported the favorable nature of the adsorption isotherm. In addition, the ‘K_f_’ values were found to be higher for nanocomposites. The trend for ‘K_f_’values was the same as observed for ‘q_m_’values obtained from the Langmuir plots, which again indicated the superiority of nanocomposite over pristine NPs. The adsorption data were also applied to the Dubinin Redushkevitch (D-R) and the Temkin isotherm models. These models and their parameters are discussed in Appendix A. The values of R^2^, ‘F’-statistics, log likelihood (G^2^) error, and χ^2^ error were calculated for all the isotherm models. The Langmuir isotherms had smaller values of χ^2^ error, ranging from 0.002 to 0.011 and a smaller G^2^ error, ranging from 0.027 to 0.031 than the other isotherm models, thus confirming the better fitting of the former model (Table 5 and Appendix A). A non-linear modeling was also applied for the best adsorbent, i.e., nanocomposite for Pb(II) adsorption, as shown in Appendix A, along with parameters given in Appendix A. Burglsser et al. reported that the non-linear behavior is more common in natural adsorbents and the response curve of column experiments in non-linear models depend on the concentration [40]. However, the batch experiments displayed better fit of linear models in the present study. In addition, G^2^ and χ^2^ errors for non-linear models were more in comparison to linear models, thus supporting the better fitting of data to later.

#### 3.3.4. Adsorption Mechanism and Desorption Studies

The adsorption mechanism involved the surface complexation and electrostatic interaction of heavy metal ions with surface functional groups. There are many oxygen (O) and nitrogen (N) functional groups present on the NGO surface. The nanocomposite has additional unsaturated charges such as M-OH and Fe-OH hydroxyl groups on the surface due to ferrite NPs in addition to N and O containing functionalities, which acted as binding sites for adsorption of metal ions. The presence of NGO increased the surface area of nanocomposite to 52.86 m^2^/g as compared to pristine NPs (23.45 m^2^/g), thus providing greater surface area for Pb(II) adsorption. Surface charge also played a very important role in the adsorption mechanism. At the optimized pH 6.0, zeta potential plots showed the more negative surface charge (−10.0 mV) for nanocomposite compared to pristine CaFe_2_O_4_ NPs (−4.9 mV) (Appendix A). This indicated larger interactions between positively charged metal ions and negatively charged nanocomposite for higher adsorption.

Figure 9 shows the XPS spectrum of CaFe_2_O_4_-NGO nanocomposite before and after Pb(II) adsorption. The characteristic peaks of C, N, O, Ca, and Fe were observed in the XPS spectrum of bare CaFe_2_O_4_-NGO nanocomposite (Figure 9a). The ‘C1s’ XPS displayed the peak at 280.1 eV, which was assigned to the C-C bond with sp^2^ hybridization. This suggested the presence of graphite-like carbon. The peak at binding energies of 345.5 eV and 400.2 eV corresponded to the ‘Ca2p’and ‘N1s’, respectively [41,42]. The peak at 528.5 eV is characteristic of oxygen due to metal-oxide bond, which suggested the presence of functional groups containing oxygen. The binding energies of ‘C1s’and ‘O1s’ in XPS spectrum of Pb(II) adsorbed CaFe_2_O_4_-NGO exhibited a slight positive shift as compared to that of CaFe_2_O_4_-NGO before adsorption (Figure 9b). This indicated a change in the local binding environment, as the electrostatic forces may get disturbed due to the metal binding. Similar results were observed by Kaur et al. for the MgFe_2_O_4_-GO nanocomposite for sequestration of Pb(II) and Ni(II) ions [20]. After adsorption, the peak of ‘Pb4f’ appeared at the binding energies of 139.8 eV confirming the presence of Pb^2+^ ions on nanocomposite.

Desorption and reuse of the adsorbents may reduce the cost of operation and open the possibility of recovering the metals extracted from the liquid phase. Six adsorption- desorption cycles were performed to test the reusability of CaFe_2_O_4_-NGO nanocomposite as the adsorbent for removal of Pb(II) ions (Figure 9c). Initially, the removal efficiency of Pb(II) ions by CaFe_2_O_4_-NGO nanocomposite was 93.5%. After the sixth adsorption-desorption cycle, the adsorption efficiency of CaFe_2_O_4_-NGO for Pb(II) ions was measured to be ≈81.7%. The results suggested the reusability of nanocomposite during the adsorption of Pb(II) ions.

#### 3.3.5. Effect of Co-Existing Ions

The influence of common existing cations on Pb(II) adsorption (Appendix A) showed that the co-existing ions in the solution significantly affected the removal of Pb(II) ions in Pb-Cd-Zn-Ni multi-ion system. CaFe_2_O_4_-NGO nanocomposite followed the metal ion removal trend as: Pb(II) > Cd(II) > Zn(II) > Ni(II). This trend could be explained on the basis of the ionic radii of metal ions. As the hydration capacity of ion decreases with an increase in ionic radii, it results in smaller hydration of ionic radii and hence higher adsorption [43]. The effect of anions on the removal of Pb(II) using synthesized nanocomposite (Appendix A) displayed the trend as Cl^−^ > NO_3_^−^ > SO_4_^2−^ using different salts. This could be attributed to the formation of ion-pairs between ligands and heavy metals [20]. The higher the stability of the ion-pair, the lesser is the availability of Pb(II) ions for adsorption. Herein, SO_4_^2−^ forms most stable ion pair with Pb(II) ions over NO_3_^−^ and Cl^−^ ions resulting in lesser adsorption of Pb(II) from PbSO_4_ solution.

## 4. Adsorption Studies for Congo Red and *p*-Nitrophenol

The effect of contact time on the removal of congo red and *p*-nitrophenol using CaFe_2_O_4_ NPs and CaFe_2_O_4_-NGO nanocomposite is displayed in Figure 10a,b. For congo red, the adsorption rate increased rapidly up to 60 min, after which the rate was slowed, and the complete equilibrium was attained in 120 min. The CaFe_2_O_4_-NGO nanocomposite and CaFe_2_O_4_ NPs exhibited the removal efficiency of 88.0% and 74.5% in 120 min, respectively. Similarly for *p*-nitrophenol, CaFe_2_O_4_-NGO nanocomposite and CaFe_2_O_4_ NPs showed the highest removal efficiency of 95.1% and 90.1% in 120 min. Thus, nanocomposites exhibited the better removal efficiency than CaFe_2_O_4_ NPs. This may be due to the presence of mesoporous N-GO in the nanocomposite, which provided more adsorption sites required for congo red and *p*-nitrophenol adsorption. The pH and adsorbent dose studies are discussed in Appendix A. Optimum pH for congo red and *p*-nitrophenol were 3.0 (Appendix A) whereas, optimum adsorbent dose for congo red and *p*-nitrophenol was 1.0 g/L and 0.1 g/L (Appendix A), respectively.

In order to study the kinetics and the adsorption mechanism, pseudo-first (Appendix A) and pseudo-second order kinetic models were applied (Figure 10c,d). The data were best fitted in pseudo-second-order model for both congo red and *p*-nitrophenol. The values for equilibrium adsorption capacities (q_e_) and rate constants calculated from these plots are reported in Table 6. The calculated and experimental q_e_ values were in agreement for pseudo-second order model. Coefficients of determination (R^2^) for pseudo-second order kinetic model were 0.99 for both the adsorbents whereas the values for pseudo-first order model ranged from 0.96 to 0.98, which indicated that the adsorption kinetics obeyed the pseudo-second order kinetic model. This supported that the concentrations of both the adsorbates and adsorbents were involved in the rate determination of adsorption. The fitting quality of model to the experimental data was determined from the values of ‘F’- statistics, SSE and χ^2^ error. The values of these parameters are summarized in Table 6. The higher values of ‘F’- statistics ranging from 44,923.0 to 77,169.0 and the smaller values of SSE and χ^2^ for pseudo-second order model were compared to the pseudo-first order model, indicating a better fitting of the former model.

## 5. Photocatalytic Degradation Studies

The photocatalytic potential of CaFe_2_O_4_-NGO nanocomposite for congo red and *p*-nitrophenol was examined under visible light irradiation. The temporal UV-Visible spectra of degraded congo red and *p*-nitrophenol solutions are illustrated in Figure 11a,b. The absorbance of congo red and *p*-nitrophenol decreased as the irradiation time increased, thus indicating their degradation. The gradual decrease of absorption peak at 498 and 398 nm suggested that the congo red and *p*-nitrophenol were removed slowly with an increase in the degradation time. The absorption maxima diminished with the progress of reaction and colored solutions changed to colorless with complete degradation of *p*-nitrophenol. The formation of a new peak at 290 nm further supported its degradation to hydroquinones [44], where the highly oxidizing capacity of ^●^OH radicals induced by the N-GO resulted in the hydroquinone formation.

In order to study the photocatalytic performance of CaFe_2_O_4_-NGO, the kinetics of congo red and *p*-nitrophenol photodegradation were investigated studied using pseudo-first and pseudo-second order kinetic models. The plot of ln(q_e_−q_t_) vs. time (t) plot to obtain pseudo-first order fitting is presented in Appendix A. The t/q_t_ vs. time (t) plot for the pseudo-second order model is shown in Figure 11a-inset and Figure 11b-inset where the correlation coefficient was 1.0. The calculated and experimental q_e_ values were better in agreement in the pseudo-second order model than in the pseudo-first order model. The fitting of the model to the experimental data was determined from the statistical analysis, and is summarized in Table 7. The higher values of ‘F’- statistics and the lower value of SSE and χ^2^ error supported better fitting of the pseudo-second order model.

### 5.1. Identification of Active Species and Photocatalytic Mechanism

In order to explore the photocatalytic degradation mechanism, trapping experiments to evaluate possible reactive species were performed. Figure 12a depicts the effect of added scavengers on the photocatalytic degradation of congo red and *p*-nitrophenol by using CaFe_2_O_4_-NGO nanocomposite. The degradation of congo red dye was decreased to greater percentage on addition of butanol (^●^OH scavenger), which indicated that ^●^OH radicals may be the major species responsible for degradation. In the case of *p*-nitrophenol, butanol and ascorbic acid (O_2_^●−^ scavenger) suppressed the rate of the degradation, suggesting that the both ^●^OH and O_2_^●−^ radicals contributed to the degradation of *p*-nitrophenol [45].

The presence of NGO in the nanocomposite was observed through the XPS measurements (Figure 9a), and resulted in the significant increase in surface area from 23.45 m^2^/g to 52.86 m^2^/g for CaFe_2_O_4_-NGO. The porous surface of nanocomposite as confirmed from BET studies, supported the availability of greater surface-active sites for adsorption, followed by photocatalysis. At the working pH of *p*-nitrophenol and congo red, i.e., at pH 3.0, the positive surface of composite as observed from zeta potential studies (Appendix A) showed higher affinity for both the anionic organic pollutants.

The mechanism for the photocatalytic degradation involved the formation of heterojunction of CaFe_2_O_4_-NGO, which caused decreased bandgap of the nanocomposite. This in turn facilitated the degradation of congo red and *p*-nitrophenol using the CaFe_2_O_4_-NGO nanocomposite. When visible light was irradiated on the nanocomposite, CaFe_2_O_4_ NPs transferred its excited electrons to NGO, which itself contained the extra electron. The electrons would react with adsorbed oxygen present on the surface of adsorbent to produce superoxide radicals (O_2_^●−^). The superoxide radical further reacted with adsorbed H^+^ present on the surface to produce peroxide radical (●OOH) further yielding H_2_O_2_ and hydroxyl radicals (^●^OH). In present study, NGO played significant role in enhancing the efficiency of CaFe_2_O_4_ NPs by increasing the conductivity and by prohibiting the recombination of photogenerated electron-hole pair at CaFe_2_O_4_ surface, as shown in Figure 1.

### 5.2. Regeneration Studies

Regeneration of photocatalyst for reuse is considered as an essential factor in sustainable applications [46]. Six cycles were performed to test the reusability of CaFe_2_O_4_-NGOnanocomposite for the photocatalytic degradation of congo red and *p*-nitrophenol (Figure 12b). After each degradation cycle, the CaFe_2_O_4_-NGOnanocomposite was separated from the reaction mixture, washed with de-ionized water, dried, and reused. After sixth photocatalytic cycle, the efficiency of CaFe_2_O_4_-NGOfor congo red was observed to be 78.3%. In case of *p*-nitrophenol, 83.2% photocatalytic efficiency was retained. Thus, reusability of the nanocatalyst further supported its future application for the removal of organic contaminants from water.

### 5.3. Photocatalytic Degradation of Congo Red and p-Nitrophenol in the Presence of Pb(II) Ions

Photocatalytic experiments were performed using Pb(II) adsorbed CaFe_2_O_4_-NGO nanocomposite in a solution containing congo red and *p*-nitrophenol, separately. The concentrations of congo red and *p*-nitrophenol were kept at 2.0 mg/L. The time dependent degradation curves are shown in Appendix A. The degradation studies before and after Pb(II) adsorption, the synergistic effect was observed after adsorption of Pb(II) in case of both congo red and *p*-nitrophenol degradation. So, the removal efficiency of nanocomposite was increased during the synergistic studies. The enhanced photocatalytic activity could be described to the electronic structural properties of Pb(II) ions. The outer shell of Pb(II) (i.e., p-orbit (6p^2^)) requires a significant number of electrons and facilitates charge transfer in photocatalytic reaction due to the higher electron affinity. However, there might be possibility of photodeposition of Pb(II) on the surface of nanocomposite during irradiation, which could play a role in enhancing photogenerated charge transfer [47]. However, in case of *p*-nitrophenol degradation (Appendix A), the absorbance values were found to increase at 290 nm, which indicated the formation of large number of hydroquinones and thus the improved degradation of *p*-nitrophenol in applying Pb(II) adsorbed CaFe_2_O_4_-NGO nanocomposite. Thus, it could be inferred that the removal efficiency of nanocomposite increased during the synergistic adsorption of Pb(II) and photocatalytic degradation of congo red and *p*-nitrophenol.

## 6. Conclusions

The CaFe_2_O_4_-NGO nanocomposite was synthesized successfully by the facile ultra-sonication method. The structural, adsorption, and photocatalytic properties of the synthesized nanocomposite were compared with the pristine ferrite NPs. Nanofabrication of NGO containing CaFe_2_O_4_ increased its potential for the effective removal of Pb(II) ions and photocatalytic degradation of congo red and *p*-nitrophenol. The maximum adsorption capacity of nanocomposite for Pb(II) ions was 780.5 mg/g, which was significantly higher than that of pristine CaFe_2_O_4_ (520 mg/g). The thermodynamic studies revealed the spontaneous and endothermic nature of the adsorption process. Under visible light illuminations, 94.8% *p*-nitrophenol and 88.2% congo red degradations were observed, which signify its effectiveness in remediating organic pollutants. The efficient photocatalytic performance of nanocomposite could be related to the presence of NGO, which not only resulted in an increase in surface area but also facilitated the fast transfer of electron and lowered the recombination of photoinduced charge carriers, supported by the results of PL measurements. Hydroxyl radicals (^●^OH) and superoxide (O_2_^●−^) radicals were largely responsible for the degradation of congo red and *p*-nitrophenol. Overall, The CaFe_2_O_4_-NGO magnetic nanocomposite has high potential for the effective removal of heavy metals and organic pollutants in water.

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
