# Peer review of "Synthesis of CaFe2O4-NGO Nanocomposite for Effective Removal of Heavy Metal Ion and Photocatalytic Degradation of Organic Pollutants"

_nanomaterials, 2021, doi:10.3390/nano11061471_

Round 1

Reviewer 1 Report

This study “Synthesis of CaFe2O4-NGO nanocomposite for effective removal of heavy metal ion and photocatalytic degradation of organic pollutants” focuses on the synthesis of magnetic nanocomposite of calcium ferrite with nitrogen doped graphene oxide (CaFe2O4-NGO).

- This is a very nice publication carefully put together, well written, and well explained, with lots of work and experiments having been done, and lots of useful data. A very complete manuscript. Only part 5 is a little bit confusing since it is not clear why “Regeneration studies” is reported. and how this is related to the Photocatalytic degradation. Also, figure 11 seems to be confusing since it is the photogenerated electrons were combined with nitrite ion. Please clarify.

- The language used in the introduction can be more specific to the scope and aim of the study. English grammar should be double-checked.

- The methodology section is not well organized for the readers to understand the concept.

- Illustrate better the Optical Studies and the results obtained.

- Quality of figures 4-8 is poor, should be improved.

Author Response

 Thanks for the sugesstions,The replies are attached 

Reviewer 2 Report

The article by Kaur and co-workers provides a detailed description of a nanocomposite material for organic pollutant removal. A good amount of new data is presented. Overall the manuscript is of good scientific quality. The topic fits well the journal’s scope. However, there are multiple points to be addressed.

1. The investigated concentration range was 0.1-2 g/L, which is much higher than practically relevant. The authors should provide some reference what is the contamination level of the pollutants in real samples.

2. The quality and organization of the figures is weak. More care should be taken to present the results and facilitate understanding of the work.

3. How were the error bars derived? Description about the derivation of errors should be included. How many independently prepared materials were tested?

4. The authors claim increased surface roughness by SEM analysis (Figure 2) but SEM does not provide surface roughness data. AFM should be performed if roughness analysis is necessary.

5. The wide application of graphene nanocomposites for the treatment of environmental pollutants should be briefly acknowledged in the introduction (10.1016/j.apmt.2020.100878; 10.1016/j.cej.2020.124642; 10.1016/j.cej.2020.126647).

6. Figure 9 should have a 2nd y axis with the adsorption capacity values for each cycle (mg Pb per g material).

7. Scheme 1 does not have much information content that needs to be represented in a figure format, the experimental procedure is simple, common and can be described in the text only.

8. The purity and grade of all chemicals, materials and solvents used in the study should be given under the materials sections.

9. Both the quotient (“x/y”) and negative exponent (“x y-1”) formats are used in the manuscript for units. Either of them should be used consistently, preferably the negative exponent format, which is recommended by the IUPAC.

10. The conclusion section should have the main results in quantitative statements as well.

Author Response

Thanks for the suggestions.The replies are attached

Reviewer 3 Report

The manuscript is very interesting; however, due to the amount of data and images, it is very difficult to read.

The novelty of the paper should be underlined in the last paragraph of the introduction, in a more precise manner.

Results and discussion. In current form, the level of section is weak to moderate and the manuscript seems to be only an enumeration of information/ obtained results. This issue should be corrected by highlighting the main findings, shortening the information presented, while keeping only the significant results and compare them with similar studies. Indeed, there is a lot of work and results, but in this form, the manuscript is difficult to read and to remark the most important findings. The authors should consider moving some information in SI file.

Formatting and quality of the figures should be improved (the plots are not of the same size, some of them are too large).

Figure 1 is not appropriate for a research article.

Please use a space between the numerical value and unit symbol (0.01-2.0g/L, line 172) should be written (0.01-2.0 g/L.

Certainly, there are impressive amount of results. However, the (too long) conclusions section needs to improve with selected and highlighted main findings.

The lines 710-740 should be customized for the current manuscript.

I consider that the article can be accepted for publication only after a major revision.

Author Response

(The authors gave the same response as above.)

Round 2

Reviewer 1 Report

Some questions and problems have been fixed by the authors, but the authors must solve the problems in the manuscript. in part of characterization, the tools and experimental devices and methods should be described.

Author Response

Thanks the reply is attached

Best regards

Reviewer 2 Report

The manuscript has been sufficiently improved.

Author Response

Thanks for the positive response

Best regards

Reviewer 3 Report

It is unacceptable to pretend that corrections were made when they were not. The following comments were discarded:

Please use a space between the numerical value and unit symbol. Example: 140.1mg/g, etc.

Chapter 2.1.4. Characterization. Please describe equipment used in the experiment – work development environment / work apparatus should be given – model of equipment (manufacturer, city, country).

If the manuscript will not be revised accordingly, I will not recommend its publication.

Author Response

Thanks for the suggestions

The replies are attached

Best regards

Round 3

Reviewer 1 Report

The article can be accepted for publication.

Author Response

Thanks for your positive response

Best regards

Reviewer 3 Report

1. Again, please use a space between the numerical value and unit symbol during the entire manuscript.
 Example: 143.0mg/g, 30min, 100℃, etc.

2. Please correct the chemical formula, if case. Example: replace "Pb(NO3).4H2O" by "Pb(NO3)·4H2O".

3. Figure 1. Please use "Intensity (a.u.)" on x-axis scale. Also, please corelate the figure with the Characterization section where you stated that the XRD measurements were performed in 2θ range from 5 to 80o.

4. Again, formatting and quality of the figures should be improved (the plots are not of the same size, some of them are too large). Example: Figure 5 a, b and c.

Author Response

Thanks for the suggestions we have incorporated the suggestions